# Chemical Analyses and Antimicrobial Activity of Nine Kinds of Unifloral Chinese Honeys Compared to Manuka Honey (12+ and 20+)

**DOI:** 10.3390/molecules26092778

**Published:** 2021-05-08

**Authors:** Yan-Zheng Zhang, Juan-Juan Si, Shan-Shan Li, Guo-Zhi Zhang, Shuai Wang, Huo-Qing Zheng, Fu-Liang Hu

**Affiliations:** College of Animal Science, Zhejiang University, No. 866, Yuhangtang Road, Xihu District, Hangzhou 310058, China; 21417023@zju.edu.cn (Y.-Z.Z.); 18868108834@163.com (J.-J.S.); lishanshan@zju.edu.cn (S.-S.L.); zhangguozhi@zju.edu.cn (G.-Z.Z.); troywang0420@foxmail.com (S.W.); hqzheng@zju.edu.cn (H.-Q.Z.)

**Keywords:** Chinese honey, antimicrobial activity, Fennel honey, Agastache honey, Pomegranate honey, volatile profile

## Abstract

Honey has good antimicrobial properties and can be used for medical treatment. The antimicrobial properties of unifloral honey varieties are different. In this study, we evaluated the antimicrobial and antioxidant activities of nine kinds of Chinese monofloral honeys. In addition, headspace gas chromatography-ion mobility spectrometry (HS-GC-IMS) technology was used to detect their volatile components. The relevant results are as follows: 1. The agar diffusion test showed that the diameter of inhibition zone against *Staphylococcus aureus* of Fennel honey (21.50 ± 0.41 mm), Agastache honey (20.74 ± 0.37 mm), and Pomegranate honey (18.16 ± 0.11 mm) was larger than that of Manuka 12+ honey (14.27 ± 0.10 mm) and Manuka 20+ honey (16.52 ± 0.12 mm). The antimicrobial activity of Chinese honey depends on hydrogen peroxide. 2. The total antioxidant capacity of Fennel honey, Agastache honey, and Pomegranate honey was higher than that of other Chinese honeys. There was a significant positive correlation between the total antioxidant capacity and the total phenol content of Chinese honey (r = 0.958). The correlation coefficient between the chroma value of Chinese honey and the total antioxidant and the diameter of inhibition zone was 0.940 and 0.746, respectively. The analyzed dark honeys had better antimicrobial and antioxidant activities. 3. There were significant differences in volatile components among Fennel honey, Agastache honey, Pomegranate honey, and Manuka honey. Hexanal-D and Heptanol were the characteristic components of Fennel honey and Pomegranate honey, respectively. Ethyl 2-methylbutyrate and 3-methylpentanoic acids were the unique compounds of Agastache honey. The flavor fingerprints of the honey samples from different plants can be successfully built using HS-GC-IMS and principal component analysis (PCA) based on their volatile compounds. Fennel honey, Agastache honey, and Pomegranate honey are Chinese honey varieties with excellent antimicrobial properties, and have the potential to be developed into medical grade honey.

## 1. Introduction

Honey is a health care product with high nutritional value, and it is also a traditional medicine that prevents and treats wound infections [1]. In recent decades, the extensive use of antibiotics has brought about bacterial resistance, and the emergence of super bacteria has caused people concern, the call for returning to traditional antimicrobial drugs is increasing. As a representative of traditional antimicrobial remedies, honey has received more and more attention, and there is more and more research on its antimicrobial properties [2].

The botanical origin of honey is important because it can significantly affect the phytochemicals present and can consequently impact on the antimicrobial capacity [3]. Many in vitro and in vivo studies have demonstrated the antimicrobial, antifungal, and antiviral activity of honey [4,5,6]. The antimicrobial ability of honey is the result of a combination of many factors, which is also believed to be the reason that honey does not produce bacterial resistance. The high osmotic pressure and low pH of honey are not conducive to the growth of microorganisms. Honey still has antimicrobial properties after being diluted, indicating that it contains other antimicrobial active compounds. Hydrogen peroxide, methylglyoxal (MGO), and polyphenolic compounds were considered to be the key compounds for honey to exert antimicrobial activity [3].

Hydrogen peroxide is the main component of honey to exert peroxide antimicrobial activity, and it is also the first antimicrobial substance identified in honey [7]. Hydrogen peroxide is mainly produced by glucose oxidase incorporated into nectar by bees, and its function may be to prevent the fermentation of immature honey [8]. When mature honey is diluted to 30–50% of the original concentration, the hydrogen peroxide has the highest accumulation rate [1]. The content of hydrogen peroxide in honey is different, and hydrogen peroxide is the main antimicrobial component of most honey [9].

After neutralizing the hydrogen peroxide in honey, most honey will lose its antimicrobial activity. However, the antimicrobial activity of Manuka honey was not affected. The pronounced non-hydrogen peroxide antimicrobial activity of Manuka honey directly originated from MGO and its concentration in Manuka honey ranged from 38 mg/kg to 761 mg/kg, which was up to 100-fold higher compared to ordinary honeys [10]. The correlation between the content of MGO in Manuka honey and its non-hydrogen peroxide antimicrobial activity was as high as 0.98 [11]. The content of MGO in Manuka honey is an important basis for its grading.

Polyphenols, as secondary metabolites of plants, are the main compounds in honey that promote human health [12]. Polyphenols can scavenge free radicals and inhibit oxidation, and a variety of polyphenolic compounds in honey have been identified as having antimicrobial activity [13,14]. The polyphenols composition in honey mainly depends on its floral origin, and it can in fact also be used as a tool for authentication of the botanical origin of honey, especially in the case of unifloral varieties [15].

The aroma of honey is specific due to the combination of volatile compounds present in low concentrations. Honey volatiles can be used as a fingerprint for the botanical origin of honey [16]. Some of the aroma compounds exhibit antimicrobial properties. The volatile components in Hungarian honeys have antimicrobial effects, and their content is 0.12–0.26% [17]. Volatile and non-volatile/semi-volatile compounds may be contributed partly to the antimicrobial activity of Ulmo honey [18].

The antimicrobial and antioxidant capacity of honey has been widely confirmed. It has been developed into medical treatment in the form of medical grade honey. As a representative of special honey, Manuka honey has been recognized by consumers for its superior antimicrobial activity and is a world-renowned medical grade honey. China is a big beekeeping country and has a wealth of nectar plant species. The number of bee colonies and volume of honey production have long been ranked first in the world. Are there any honey varieties in China that have antimicrobial properties close to or exceeding Manuka honey and what is their antimicrobial mechanism? In order to solve this question, we researched nine species of unifloral Chinese honeys, hoping to find Chinese honey varieties with excellent antimicrobial ability.

## 2. Results and Discussion

### 2.1. Conventional Physical and Chemical Indexes

In order to ensure that the botanical origin of unifloral honey samples was authentic, we commissioned trusted beekeepers to sample at designated locations during the corresponding flowering period. The information of nine Chinese unifloral honey samples was shown in Table 1. We tested the physicochemical properties of the honey samples, and the relevant results are shown in Table 2. To further determine the purity of the samples, we measured the pollen ratio of the honey samples. The corresponding pollen percentages of Chinese monofloral honey samples were all greater than or equal to 75.00 ± 1.50%, which indicated that the purity of our monofloral honeys was high.

The water contents for seven Chinese honeys were over 20%. This may be related to the production mode of honey in China. In order to pursue the yield or the purity of unifloral honey, many Chinese honeys do not have enough time to mature in beehives. The diastase activity of Manuka honey was significantly lower than that of Chinese honey. This may be related to the high dose of MGO in Manuka honey. MGO has negative effects on the structure and function of proteins in Manuka honey [19]. The values of HMF for Manuka honeys were quite high in comparison with the rest of the honeys. Manuka honeys were purchased from New Zealand Comvita Ltd. (Wellington, New Zealand), which had been stored at room temperature for a longer time before sampling than other samples. In addition, the possible heating treatment in the factory may have also increased the content of HMF in Manuka honeys. Chinese honey samples were all raw honey, which were directly stored in the refrigerator after being taken out from the hives.

The chroma value of Manuka honey 20+ was as high as 150 ± 1.00 mm. Fennel honey and Agastache honey have chroma values of 82 ± 0.00 mm and 71 ± 0.00 mm respectively, which were significantly higher than other Chinese honeys. The chroma value was positively correlated with total antioxidant capacity and inhibition zone diameter. The correlation between the chroma value of Chinese honeys and the total antioxidant capacity was 0.940. The correlation between the chroma value with the inhibition zone diameter was 0.746. Dark honey contained a lot of antioxidant components, and the antimicrobial ability of dark honey was generally stronger than that of light honey [20,21]. Our research showed that darker honeys provide high levels of antioxidants as well as antimicrobial effectiveness.

### 2.2. Antimicrobial Test

The antimicrobial activity of honey samples against *Staphylococcus aureus* (gram-positive)*, Escherichia coli* (gram-negative), and *Candida albicans* (fungus) was determined by the agar diffusion and broth macrodilution method. These three microorganisms were the recommended species for the antimicrobial test in the Chinese disinfection technical specifications. 10% phenol and Manuka honey were used as positive control. The negative control was distilled water. The agar-well diffusion test used with 50% solution of honey. All honey samples had no bacteriostatic ring against *E. coli* and *C. albicans*. As for *S. aureus*, nine Chinese honey samples had a bacteriostatic ring larger than 11 mm in diameter (Table 3). The difference in the sensitivity of gram-positive bacteria and gram-negative bacteria to honey is a result of the difference in the composition of their cell walls. Compared with gram-negative bacteria, gram-positive bacteria have no outer membrane to protect the peptidoglycan layer, which makes it easier for the antimicrobial agents to penetrate and cause damage [22]. Manuka honey has been proven to inhibit the gram-positive bacterium *Enterococcus faecalis*, while gram-negative *E. coli* was more resistant to it [23]. The biofilm of *C. albicans* is special and can switch from a yeast form to a filamentous form. This increases the difficulty of suppressing it [24]. The bacteriostatic ring diameter of Fennel honey (21.50 ± 0.41 mm), Agastache honey (20.74 ± 0.37 mm), and Pomegranate honey (18.16 ± 0.11 mm) against *S. aureus* was significantly larger than other honeys, including Manuka honey 12+ (14.27 ± 0.10 mm) and Manuka honey 20+ (16.52 ± 0.12 mm). The artificial honey sample had no bacteriostatic ring, indicating that the antimicrobial effect was not simply conditional to the sugar content in the honey. It is well known that the agar-well diffusion method is suitable for the detection of antimicrobial effect, but if the minimum inhibitory concentration (MIC) and minimum bactericidal concentration (MBC) are to be calculated, a more sensitive method, such as broth dilution method, must be used. By this method, the microorganisms are in direct contact with the testing inhibiting substance. The antimicrobial effect does not depend on the diffusibility of the reagent through the agar medium [25].

Broth micro-dilution method was used to detect MIC and MBC of the 11 honey samples with obvious bacteriostatic ring by agar-well diffusion test. 16 h, 12 h, and 18 h were the rapid growth periods of K/2 of *S. aureus*, *E. coli,* and *C. albicans*, respectively, which were determined as the determination time of MIC experiment. For each honey sample, ten concentration gradients were prepared for this test. We found that the inhibition rate was not linearly related to the concentration. The result of broth dilution assay (Table 4) showed that the inhibitory effect of honey on *S. aureus* was better than that of *E. coli* and *C. albicans*, which was consistent with the result of the agar-well diffusion test. The MIC of honey samples on *E. coli* were all significantly lower than that on *C. albicans.* This was consistent with the report that *C. albicans* was difficult to inhibit [24]. The MBC of honey samples was relatively high, close to 50%, which is related to the low concentration of bactericidal substances in honey. The antimicrobial ability of honey is the result of a combination of many compounds [1]. The MBC of Fennel honey (25.0–40.0%) against *S. aureus* was lower than that of other honeys, including Manuka honey (>50.0%). Fennel honey, Agastache honey, and Pomegranate honey were three kinds of Chinese honeys with excellent antimicrobial effect. The results of MIC and MBC also showed that honey has a better inhibitory effect on gram-positive bacteria.

### 2.3. Analysis of Antimicrobial Components in Honey

In order to study the antimicrobial mechanism of Chinese honey, we analyzed the antimicrobial components of the above 11 honey samples. Hydrogen peroxide is considered to be the main antimicrobial substance in honey [7]. After adding catalase, the antimicrobial effect of all nine Chinese honey samples was abolished, while the two Manuka honey samples still maintained a significant antimicrobial effect on *S. aureus.* MGO was identified as the main antimicrobial component of Manuka honey [26]. We tested the hydrogen peroxide and MGO content of the honey samples. Hydrogen peroxide concentrations ranged from 7.31 ± 0.72 µm/g to 2882.76 ± 10.86 µm/g in different honey samples (Table 3). The content of hydrogen peroxide in Chinese honey was significantly higher than that in Manuka honey. Fennel honey, Agastache honey, and Pomegranate honey produced 2882.76 ± 10.86 µm/g, 1161.14 ± 4.70 µm/g and 2150.89 ± 32.30 µm/g of hydrogen peroxide, whereas Manuka 12+ (92.26 ± 0.13 µm/g) and Manuka 20+ (7.31 ± 0.72 µm/g) produced low amounts of hydrogen peroxide. This result can be attributed to the observation that Manuka honey lacks hydrogen peroxide [5].

The MGO content of Manuka 12+ and 20+ was 109.95 ± 0.23 mg/kg and 268.27 ± 0.35 mg/kg, respectively, which is not significantly different from the content on the label. MGO was not detected (below LOD) in all nine Chinese honey samples. The artificial honey sample had no antimicrobial activity, which indicates that the antimicrobial effect is not simply conditional to the sugar content in the honey. Hydrogen peroxide may be the main antimicrobial factor of Chinese honey samples. The correlation coefficient between the hydrogen peroxide content and the diameter of the inhibition zone in Chinese honey was 0.535, indicating that its antimicrobial activity was the result of the combined action of multiple bacteriostatic components. These results clearly indicate that factors other than hydrogen peroxide production, for example, polyphenolic and volatile compounds, contribute to the antimicrobial activity. Therefore, we identified polyphenolic and volatile compounds in the honeys and their possible relation to the antimicrobial activity was assessed.

Polyphenolic compounds have been recognized as the main reason for the antioxidant activity of honey. Many polyphenolic compounds in honey have also been identified as having antimicrobial activity [3]. The content of total phenols and total flavonoids in honey and the antioxidant capacity of honey were measured (Table 5). The correlation of these indicators was analyzed (Table 6). The total phenol content of Fennel honey (81.27± 1.31 mg/100 g, GAE) was between Manuka 12+ (66.14 ± 0.92 mg/100 g, GAE) and Manuka 20+ (83.54 ± 0.23 mg/100 g, GAE), which was significantly higher than that of other Chinese honeys. Alzahrani [20] reported that Manuka honey had the highest phenolic content (899.09 ± 11.75 mg/kg, GAE) among the four floral honeys examined in their research. Our research results were essentially the same. The total antioxidant capacity of Fennel honey (2601.84 ± 51.23 μg/100 g, Rutin) was higher than other Chinese honeys. Pomegranate honey (48.88 ± 0.27 mg/100 g, QE) has the highest total flavonoid content in Chinese honeys. The DPPH scavenging ability of Agastache honey (0.77 ± 0.02 mg/mL) was the best among the Chinese honeys, between Manuka honey 12 + (0.69 ± 0.02 mg/mL) and Manuka honey 20 + (0.80 ± 0.03 mg/mL).

Phenolics derived from nectar are the main antioxidants of honey. Plant species have a significant impact on the antioxidant capacity of honey [27]. Many reports have shown a significant correlation between the total phenolic content and total antioxidant activity of honey [20,28]. In our research, there was a significant positive correlation between total phenolic content and total antioxidant capacity of Chinese honey (r = 0.958). The total phenol content of Chinese honey was positively correlated with the diameter of inhibition zone (r = 0.795). The phenolic compounds in honey may play a certain antimicrobial effect. Flavone content was positively correlated with DPPH scavenging power (r = 0.846). The correlation between total antioxidant capacity and inhibition zone diameter of Chinese honey was 0.853. The antimicrobial ability of honey can be predicted by its total antioxidant capacity.

### 2.4. Volatile Compounds in Honey

Some aroma compounds exhibit antimicrobial properties [17,18]. Volatile compounds have great importance in characterizing honey’s botanical source, which directly influences their sensory characteristics [16]. Headspace gas chromatography-ion mobility spectrometry (HS-GC-IMS), with no complex pretreatment and high sensitivity, has been widely used in food volatile components analysis [29].

We detected the volatile components of the three kinds of honey (Fennel honey, Agastache honey, and Pomegranate honey), and their antimicrobial properties were higher than that of Manuka honey. The pollen percentage of Manuka 12+ honey (50.91 ± 1.84%) was higher than that of Manuka 20+ honey (30.67 ± 0.81%). Therefore, Manuka 12+ honey was used as the control sample for volatile components detection. The volatile compounds in Fennel honey, Agastache honey, Pomegranate honey, and Manuka 12+ honey samples were isolated by HS and identified by GC-IMS. A total of 81 volatile components were detected, and 60 of them were identified (Table 7). Many volatiles identified in our study were aromatic compounds. Volatile benzene derivatives in honey were considered to have antimicrobial effects [30]. Volatile compounds potentially partly contributed to the antimicrobial activity of honey.

The two-dimensional (2D) array full-size top view plot (Figure 1) of HS-GC-IMS was obtained by the normalization of the ion migration time and reactive ion peak (RIP) position. The ion signal was described by different colors. Taking Fennel honey as a reference, red means high and blue means low. Increasing color darkness indicates increasing content. We used the Gallery Plot plug-in to draw the fingerprint spectrum of the volatile compounds of honey samples (Figure 2).

There were significant differences in the composition and content of volatile components among honey samples. Manuka honey had the most volatile components. Citral, 1-phenylethanol, (*Z*)-2-octenal, (*E*)-3-Octen-2-one, γ-Octalactone, and Citronellol could be used as the characteristic components of Manuka honey, which was different from the other three kinds of honey. Hexanal-D was the characteristic component of Fennel honey. 3-Methylpentanoic acid and ethyl 2-methylbutyrate were the characteristic components of Agastache honey. Pomegranate honey had the least variety of flavors, and heptanol was its unique compound. Among the four kinds of honey, the volatile compounds of dark honey were more than that of light-colored honey. Dark honey seemed to have more volatile components and this may be related to its stronger antimicrobial activity.

HS-GC-IMS applied with chemometrics, such as PCA (principal component analysis), could cluster honey with different floral origins. PCA could highlight the differences between samples. The PCA results of the flavor compounds in the four honeys are shown in Figure 3. Principal component 1 (PC1) expressed 52% of the variance and Principal component 2 (PC2) explained 34% of the variance. The loadings of each compound on the PCA explicitly showed that the grouping of the different unifloral honeys was mainly influenced by certain volatile compounds. PC1 and PC2 of all the samples explained 86% of the total variance at length. The PCA results indicated that the four samples occupied relatively independent spaces in the distribution map. The four kinds of honeys could be completely distinguished. HS-GC-IMS imaging coupled with PCA was a useful strategy to discriminate honey from different floral origins.

The botanical origin is the major factor that determines the physicochemical properties of honey, due to the wide variability in the chemical structure of plant nectars and secretions. Geographical origins, climatic conditions, honeybee associated factors, and honey’s associated factors will also affect the quality of honey to a certain extent [31]. Further testing is merited to use a statistically significant number of honeys from a certain single botanical origin, place of production, and extraction date.

## 3. Materials and Methods

### 3.1. Honey Samples

In order to ensure that the botanical origin of unifloral honey samples was authentic, we commissioned trusted beekeepers to sample at designated locations during the corresponding flowering period. Nine Chinese unifloral honey samples were collected (Table 1). Manuka honey 12+ and 20+ were purchased from New Zealand Comvita Ltd. (Wellington, New Zealand). The collection time of all the samples was in 2015–2017. All honey samples were stored in a refrigerator at −20 °C away from light until analyzed.

### 3.2. Main Reagents and Equipment

#### 3.2.1. Main Reagents

*S. aureus* (CICC 23926), *E. coli* (CICC 10899) and *C. albicans* (CICC 32380) were purchased from China Center of Industrial Culture Collection (Beijing, China). Catalase, methylglyoxal (HPLC grade), gallic acid, quercetin, and DPPH were purchased from Sigma-Aldrich (St. Louis, MO, USA). H_2_O_2_ quantitative analysis kit (water-compatible), LB broth, and phenol were purchased from Sangon Biotech (Shanghai, China). Nutrient agar was purchased from Sinopharm Chemical Reagent Co., Ltd. (Beijing, China).

#### 3.2.2. Main Equipments

Multiskan Sky: Thermo Scientific, Waltham, MA, USA. Shimadzu spectrophotometer UV-2550: Shimadzu Co., Ltd., Kyoto, Japan. Agilent 1200 liquid chromatograph: Agilent Technologies, Palo Alto, CA, USA. Agilent 490 gas chromatograph: Agilent Technologies, Palo Alto, CA, USA. IMS instrument: FlavourSpec^®^, Gesellschaft für Analytische Sensorsysteme mbH, Dortmund, Germany.

### 3.3. Physicochemical Analysis

The water content was determined by an Abbe refractometer (MASTER-20M, ATAGO, Tokyo, Japan). The color intensity was determined using a Pfund honey colorimeter (HI96785, HANNA, Winsockit, RI, USA). The chroma value of solid honey was determined after complete decrystallization at 50 °C Ash content was obtained by placing honey (6 g) in a crucible and heating at 550 °C overnight in a muffle furnace. The protein content was determined by Bradford method [32]. The sugars in honey samples were carried out by an HPLC (Agilent 1200) with an ELSD-detector, according to the method described by Deng [28]. The hydroxymethylfurfural (HMF) content of honey samples was determined by the method reported by Ribeiro [33]. Diastase activity was determined using the method reported by Pasias [34].

The pollen detection in honey was carried out according to the method described by the PRC national standard (GB/T 23194-2008) [35] with minor modifications. 10 g honey was added to 20 mL 40 °C distilled water. After fully dissolved, it was centrifuged at 3000× *g* for 10 min. We discarded 4/5 of the supernatant, then added 2 mL of acetic acid, mixed well, and let stand for 2 h. Centrifuged at 3000× *g* for 10 min. Discarded 1/2 of the supernatant. Added 3 mL of freshly prepared mixture of acetic anhydride and sulfuric acid (9:1, volume ratio). After mixing, the sample was bathed at 90 °C for 7 min. Centrifuged at 3000× *g* for 10 min and then discarded the supernatant. The precipitate was washed 3 times with distilled water. Centrifuged after each cleaning. The final precipitate was pollen grains. The extracted pollen grains were stored in 1 mL 50% glycerol. We took a drop of sample solution on the blood cell counting plate. We let it stand for 5 min, and observed with a microscope. Pollen plant source identification refers to the pollen morphology of Chinese plants [36]. The predominant specific pollen was counted from the entire slide or until at least 400 pollens were counted. Three parallel tests were performed on each sample.

### 3.4. Antimicrobial Activity of Honey

Three bacterial strains were used for antimicrobial assays, including *S. aureus* (Gram-positive), *E. coli* (Gram-negative), and *C. albicans* (fungus), which are 3 common types of bacteria in clinic. These 3 microorganisms were also the recommended species for the antimicrobial test in the Chinese disinfection technical specifications. In order to determine the non-hydrogen peroxide antimicrobial activity of honey, catalase was added to honey samples to remove hydrogen peroxide. Corresponding to the amount of sugar in honey, an artificial honey sample was made by diluting 10.37 g fructose, 10.00 g glucose, and 0.38 g sucrose in 4.25 mL sterile Milli-Q water. Two different methods were used to evaluate the antimicrobial activity of honey: agar-well diffusion and broth micro-dilution. All experiments were performed in triplicate.

#### 3.4.1. Agar Well Diffusion Assay

Suspensions of the bacteria were diluted in LB broth to provide a concentration of 10^7^ CFU/mL. 100 µL of this suspension was added to 30 mL of agar at 50 °C and poured into a 9 cm diameter plate. After cooling, 4 holes with a diameter of 8 mm were drilled into the plate. The agar hole was sealed with a flame at the bottom. Each well was filled with 100 µL honey solution, and then the plates were incubated at 37 °C for 18 h. The diameter of the inhibition zones was measured by vernier caliper. Throughout the agar well diffusion experiments, a solution of 50% (*w*/*v*) honey was used, unless otherwise stated.

#### 3.4.2. Broth Micro-Dilution Assay

The minimum inhibitory concentration (MIC) and minimum bactericidal concentration (MBC) of the honey samples were determined by the broth dilution method in 96-well microplates. We diluted the honey sample with LB broth. The final concentrations of diluted honey samples were 50%, 25%, 20%, 12.5%, 10%, 6.25%, 5%, 3.125%, 2.5%, 1.25%. The bacterial cultures were diluted to a final concentration of 10^6^ CFU/mL. Tested honey solutions and bacterial suspension were mixed in a ratio of nine to one. 200 µL mixed solution was dispensed into each well of the 96-well microplate. The 96-well microplates were incubated at 37 °C. *S. aureus* was cultured for 16 h, *E. coli* for 12 h and *C. albicans* for 18 h. These times were the K/2 rapid growth periods of the three bacteria. The absorbance was measured at 600 nm. The minimum concentration of honey to inhibit bacterial growth was considered to be MIC. 50 µL bacteria suspension without bacteria growth in MIC test was cultured on nutrient agar plate and incubated at 37 °C for 24 h. The minimum concentration without bacterial growth was MBC.

### 3.5. Hydrogen Peroxide Assay

Honeys were diluted to 50% and the hydrogen peroxide content was measured as described in the protocol of H_2_O_2_ quantitative analysis kit (water-compatible). This kit (Item No: C500069) was purchased from Sangon Biotech (Shanghai, China). In an acidic environment, H_2_O_2_ can oxidize Fe^2+^ to Fe^3+^. Fe^3+^ ions combine with xylenol orange molecules to form Fe^3+^-xylenol orange complexes. The complexes have a maximum absorption wavelength of 560 nm or 595 nm, and the absorption value is proportional to the concentration of H_2_O_2_. The standard curve was made by H_2_O_2_ with a known concentration gradient (1–100 μM). Experiments were performed 3 times in duplicate.

### 3.6. Determination of Methylglyoxal Concentration in Honeys

The content of methylglyoxal was analyzed according to the method of Oelschlaegel [37] with minor modifications. The processed samples were analyzed by Agilent 1200 HPLC system. The analytical column was a Kromasil reversed phase chromatographic C18 column (4.6 mm × 150 mm, 5.0 μm). The mobile phase A was 0.1% acetic acid and mobile phase B was methanol. The elution conditions were: 0–5 min, 30% B; 5–10 min, 30% B–90% B; 10–15 min, 90% B; 15–16 min, 90% B–30% B; 16–20 min, 30% B. The flow rate was 1.0 mL/min.

### 3.7. Antioxidant Test

#### 3.7.1. Total Phenolics

The total phenolics were determined by using the Folin–Ciocalteu method [38]. The honey sample (5 g) was diluted to 50 mL with distilled water and filtered through Whatman No. 1 paper. 1 mL of this solution was mixed with 1 mL Folin–Ciocalteu reagent and then mixed by vortexing. The mixed solution was treated with 5 mL of 1 M Na_2_CO_3_ solution, and then made up to 10 mL. The mixture was incubated at ambient temperature in the dark for 1.5 h. Absorbance was measured at 760 nm against an ethanol blank, and gallic acid was used as standard.

#### 3.7.2. Total Flavonoids

Honey (5 g) was diluted to 25 mL with distilled water and filtered through Whatman No. 1 paper. 1mL of this solution was mixed with 0.3 mL of 15% NaNO_3_ solution, we shook it well and let stand for 6 min. We added 0.3 mL 10% Al(NO_3_)_3_ solution, shook well and let stand for 6 min, added 4 mL of 4% NaOH solution, diluted with 50% ethanol to 10 mL, shook well and let stand for 15 min. The absorbance was measured at 510 nm, and quercetin was used as standard.

#### 3.7.3. DPPH Radical Scavenging Activity

The DPPH radical scavenging activity of honey samples was determined according to the method of Scherer [39] with minor modifications. 2.7 mL DPPH solution in methanol (0.06 mM) was mixed with 0.3 mL of 50% (*w*/*v*) honey solution. The mixture was thoroughly mixed by vortexing and we let stand for 1.5 h in the dark. Then, the absorbance was measured at 517 nm.

#### 3.7.4. Total Antioxidant Capacity

The total antioxidant capacity of honeys was determined according to the protocol of Atmani [40], with modifications. Honey (5 g) was diluted to 25 mL with distilled water and filtered through Whatman No. 1 paper. 1 mL of this solution was added to 2.5 mL of PBS (0.2 M, pH 6.6) and 2.5 mL of 1% K_3_Fe(CN)_6_ solution. This was incubated at 50 °C for 20 min. 2.5 mL 10% trichloroacetic acid was added to the mixture. Centrifuged at 5000× *g* for 5 min after standing for 10 min. We took 2.5 mL supernatant, added 2.5 mL distilled water, and 0.5 mL 0.1% FeCl_3_, shook well and let stand for 10 min. The absorbance was determined at 700 nm. The results were converted according to Rutin’s standard curve.

### 3.8. HS-GC-IMS

An Agilent 490 gas chromatograph and IMS instrument equipped with an automatic sampling device were used to detect the volatile substances in honey samples. The HS-GC-IMS analysis was performed as previously described [41] with minor modifications. Briefly, honey (2 g) was transferred into a 20 mL glass vial sealed with a silicon septum and magnetic metal crimp. The headspace bottle was incubated at 55 °C for 20 min. Then, 500 µL aliquots was automatically injected into the heated injector under splitless injection mode at 85 °C. The GC equipped with an FS-SE-54-CB-0.5 (15 m × 0.53 mm ID) column was used for separation at 60 °C. N_2_ (purity ≥ 99.99%) was used as the carried gas. The following were programmed as flows: initial flow of 2 mL/min, maintained 2 min, flow ramp up to 100 mL/min in 18 min, and maintained for 10 min. The drift tube length was 50 mm, which operated at a constant voltage of 400 V/cm. The drift tube was maintained at 45 °C under N_2_ as drift gas at 150 mL/min. The IMS cell was operated in the positive ion mode using helium as ionization source. Each spectrum had 32 scans with a grid pulse width of 100 μs, a repetition rate of 21 ms, and a sampling frequency of 150 kHz. PCA was used to analyze the differences in volatile components between samples of honey from 4 different plant sources, with mean-centered, UV scaled, and log-transformed data, before building the PCA model. LAV software version 2.2.1 (Gesellschaft fur Analytische Sensorsysteme mbH, Dortmund, Germany) was used to analyze the data.

### 3.9. Statistical Analysis

All analyses were carried out in triplicate, and the data were expressed as mean standard deviation (SD). Correlation analysis was achieved using SPSS 24.0 software (IBM SPSS Statistics; Chicago, IL, USA).

## 4. Conclusions

We researched the chemical composition and antimicrobial activity of nine unifloral Chinese honeys. The agar diffusion test and broth micro-dilution assay showed that Fennel honey, Agastache honey, and Pomegranate honey were Chinese honey varieties with excellent antimicrobial activity, compared to Manuka honeys. The antimicrobial activity of Chinese honey depends on hydrogen peroxide. There was a significant positive correlation between the total antioxidant capacity and the total phenol content of Chinese honey (r = 0.958). The correlation coefficient between the chroma value of Chinese honey and the total antioxidant and the diameter of inhibition zone was 0.940 and 0.746, respectively. The analyzed dark honeys had better antimicrobial and antioxidant activities. The flavor fingerprints of the honey samples could be successfully built using HS-GC-IMS and PCA, based on their volatile compounds. Hexanal-D and Heptanol were the characteristic components of Fennel honey and Pomegranate honey, respectively. Ethyl 2-methylbutyrate and 3-methylpentanoic acid were the unique compounds of Agastache honey. Fennel honey, Agastache honey, and Pomegranate honey have the potential to be developed into medical grade honey and deserve further study.

## Figures and Tables

**Figure 1 molecules-26-02778-f001:**
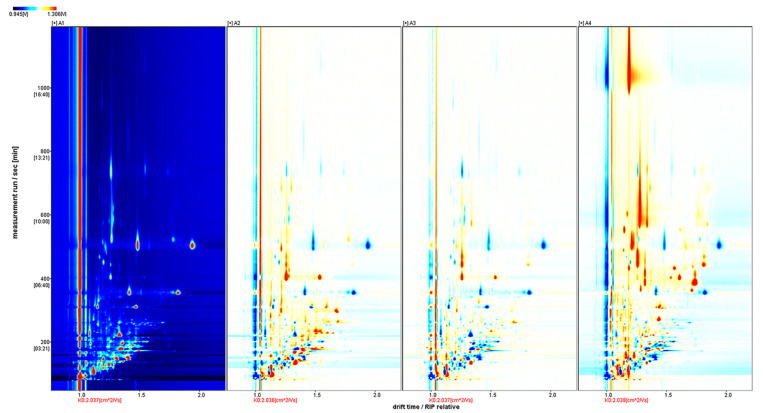
Imaging of volatile compounds represented by HS-GC-IMS of honey samples. A1: Fennel honey; A2: Agastache honey; A3: Pomegranate honey; A4: Manuka 12+ honey.

**Figure 2 molecules-26-02778-f002:**
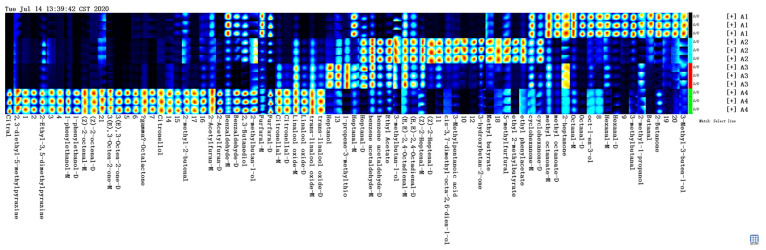
Gallery plot of the selected signal peak areas obtained with 4 kinds of honey samples. A1: Fennel honey; A2: Agastache honey; A3: Pomegranate honey; A4: Manuka 12+ honey.

**Figure 3 molecules-26-02778-f003:**
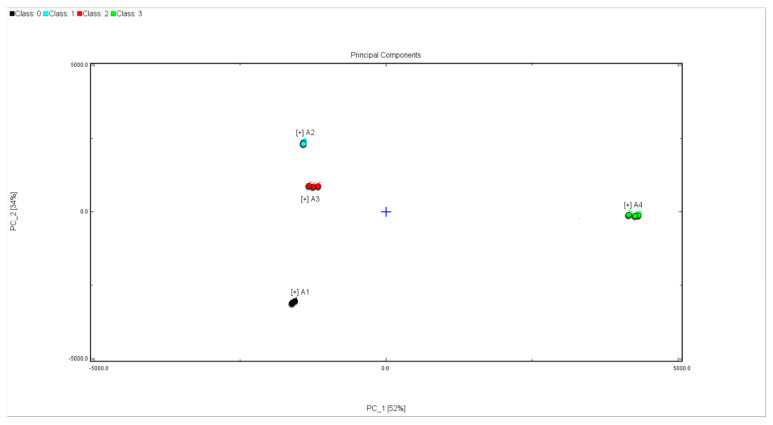
PCA scores plot of preprocessed HS-GC-IMS spectra of Fennel honey, Agastache honey, Pomegranate honey, and Manuka 12+ honey. A1: Fennel honey; A2: Agastache honey; A3: Pomegranate honey; A4: Manuka 12+ honey.

**Table 1 molecules-26-02778-t001:** The information of 9 species of unifloral Chinese honey samples.

Plant Sources (Latin Name)	Plant Sources (Common Name)	Producing Area	Sampling Time
*Zanthoxylum bungeanum* Maxim	Pepper	Zhejiang Province	2015
*Punica granatum* L.	Pomegranate	Yunnan Province	2016
*Brassica napus* L.	Rape	Jiangsu Province	2016
*Eriobotrya japonica* (Thunb.) Lindl	Loquat	Zhejiang Province	2017
*Vitex negundo* L.	Vitex	Shaanxi Province	2017
*Crataegus pinnatifida* Bunge	Hawthorn	Shaanxi Province	2017
*Helianthus annuus* L.	Sunflower	Shaanxi Province	2017
*Agastache rugosa* (Fisch. et Mey.) O. Ktze	Agastache	Yunnan Province	2017
*Foeniculum vulgare* Mill	Fennel	Gansu Province	2017

**Table 2 molecules-26-02778-t002:** Values of physicochemical parameters of the different varieties of honey.

Honey	Pollen (%)	Water Content (%)	Diastase Activity (DN)	Chroma Value (mm)	HMF (mg/kg)	Protein (%)	Ash Content (%)	Fructose (%)	Glucose (%)
Manuka 20+	30.67 ± 0.81	17.00 ± 0.15	7.22 ± 0.18	150 ± 1.00	27.87 ± 0.35	0.31 ± 0.02	0.36 ± 0.03	40.12 ± 0.75	31.02 ± 1.35.
Manuka 12+	50.91 ± 1.84	16.84 ± 0.27	9.12 ± 0.08	150 ± 0.00	33.54 ± 0.14	0.29 ± 0.02	0.42 ± 0.02	36.06 ± 0.58	30.51 ± 1.12
Fennel	78.33 ± 0.86	20.56 ± 0.12	27.12 ± 0.10	82 ± 0.00	4.62 ± 0.07	0.83 ± 0.03	0.24 ± 0.01	38.14 ± 0.84	22.86 ± 0.13
Agastache	80.36 ± 1.27	20.68 ± 0.23	21.58 ± 0.07	71 ± 0.00	2.44 ± 0.10	0.35 ± 0.02	0.37 ± 0.01	34.66 ± 0.57	28.94 ± 0.16
Pomegranate	84.94 ± 1.10	17.88 ± 0.35	21.96 ± 0.19	43 ± 1.00	1.97 ± 0.08	0.31 ± 0.01	0.25 ± 0.03	36.49 ± 0.74	34.12 ± 0.50
Hawthorn	75.00 ± 1.50	20.56 ± 0.14	22.22 ± 0.08	66 ± 1.00	2.12 ± 0.03	0.47 ± 0.02	0.15 ± 0.01	34.47 ± 0.17	28.68 ± 0.28
Pepper	83.87 ± 1.19	20.04 ± 0.35	23.01 ± 0.12	44 ± 0.00	12.18 ± 0.03	0.33 ± 0.03	0.22 ± 0.01	35.55 ± 0.41	34.63 ± 0.23
Sunflower	85.26 ± 1.16	23.32 ± 0.09	12.70 ± 0.22	21 ± 1.00	1.02 ± 0.06	0.23 ± 0.03	0.16 ± 0.01	36.30 ± 0.43	33.92 ± 0.13
Loquat	93.94 ± 1.20	22.00 ± 0.26	13.93 ± 0.08	3 ± 0.00	1.11 ± 0.06	0.14 ± 0.01	0.06 ± 0.01	34.22 ± 0.30	29.36 ± 0.14
Rape	95.24 ± 1.53	18.76 ± 0.08	22.94 ± 0.05	24 ± 0.00	1.98 ± 0.04	0.31 ± 0.02	0.08 ± 0.01	37.78 ± 0.57	35.22 ± 0.29
Vitex	90.40 ± 1.21	21.16 ± 0.10	22.12 ± 0.07	11 ± 0.00	0.90 ± 0.07	0.30 ± 0.01	0.05 ± 0.01	39.28 ± 0.65	28.75 ± 0.22

Data represent the mean of triplicate readings ± standard deviations (SD).

**Table 3 molecules-26-02778-t003:** The antimicrobial activity and main antimicrobial substance content of honey samples.

Plant Sources	Inhibition Diameters (mm)	H_2_O_2_ (μm/g)	MGO (mg/kg)
Fennel	21.50 ± 0.41	2882.76 ± 10.86	-
Agastache	20.74 ± 0.37	1161.14 ± 4.70	-
Pomegranate	18.16 ± 0.11	2150.89 ± 32.30	-
Manuka 20+	16.52 ± 0.12	7.31 ± 0.72	268.27 ± 0.35
Manuka12+	14.27 ± 0.10	92.26 ± 0.13	109.95 ± 0.23
Vitex	13.41 ± 0.25	2351.31 ± 5.43	-
Pepper	12.73 ± 0.49	428.15 ± 1.03	-
Hawthorn	12.69 ± 0.76	1579.83 ± 3.70	-
Sunflower	12.46 ± 0.98	144.24 ± 0.90	-
Rape	12.07 ± 0.17	1534.04 ± 3.20	-
Loquat	11.57 ± 0.27	966.82 ± 10.86	-
Phenol	31.39 ± 0.15	-	-

The data of inhibition zone were obtained from the agar diffusion test against *S. aureus*. Data represent the mean of triplicate readings ± standard deviations (SD)–means below LOD.

**Table 4 molecules-26-02778-t004:** Minimum inhibitory concentrations and minimum bactericidal concentrations of honey samples. The antimicrobial activity of honey was assessed against *S. aureus*, *E. coli,* and *C. albicans.*

Honey	*S. aureus*				*E. coli*		*C. albicans*		
	MIC_90_	MIC_50_	MBC	MIC_90_	MIC_50_	MBC	MIC_90_	MIC_50_	MBC
Manuka 20+	2.5–3.1	1.4–2.5	>50.0	3.1–5.0	3.1–5.0	>50.0	>50.0	25.0–50.0	>50.0
Manuka 12+	2.5–3.1	1.4–2.5	>50.0	3.1–5.0	3.1–5.0	>50.0	>50.0	25.0–50.0	>50.0
Pomegranate	2.5–3.1	2.5–3.1	>50.0	12.5–20.0	12.5–20.0	>50.0	25.0–50.0	25.0–50.0	>50.0
Fennel	5.0–6.3	5.0–6.3	25.0–40.0	>50.0	33.3–50.0	25.0–40.0	>50.0	12.5–20.0	40.0–50.0
Vitex	6.3–10.0	6.3–10.0	>50.0	12.5–20.0	12.5–20.0	25.0–50.0	25.0–50.0	20.0–25.0	25.0–50.0
Hawthorn	6.3–10.0	6.3–10.0	>50.0	12.5–20.0	12.5–20.0	25.0–50.0	25.0–50.0	25.0–50.0	>50.0
Rape	6.3–10.0	6.3–10.0	>50.0	12.5–20.0	12.5–20.0	25.0–50.0	25.0–50.0	25.0–50.0	>50.0
Agastache	10.0–12.5	5.0–6.3	>50.0	12.5–20.0	10.0–12.5	25.0–50.0	>50.0	25.0–50.0	>50.0
Sunflower	10.0–12.5	6.3–10.0	>50.0	12.5–20.0	12.5–20.0	>50.0	25.0–50.0	25.0–50.0	>50.0
Pepper	10.0–12.5	6.3–10.0	>50.0	20.0–25.0	6.3–10.0	>50.0	25.0–50.0	25.0–50.0	>50.0
Loquat	20.0–25.0	20.0–25.0	>50.0	25.0–40.0	12.5–20.0	>50.0	>50.0	25.0–50.0	>50.0

MIC90: 90% minimum inhibitory concentration. MIC50: 50% minimum inhibitory concentration. The concentration was the percentage ratio of mass to volume (*w*/*v*, %).

**Table 5 molecules-26-02778-t005:** The antioxidant activity, total phenols, and total flavonoids content of tested honeys.

Honey	Total Phenols	Total Flavonoids	Radical Scavenging Activity (DPPH)	Total Antioxidant Capacity
	(mg/100 g, GAE)	(mg/100 g, QE)	(IC50, mg/mL)	(μg/100 g, rutin)
Manuka 20+	83.54 ± 0.23	60.72 ± 0.29	0.80 ± 0.03	4224.17 ± 21.39
Manuka 12+	66.14 ± 0.92	50.15 ± 0.12	0.69 ± 0.02	3091.97 ± 27.66
Fennel	81.27± 1.31	31.81 ± 0.08	0.23 ± 0.18	2601.84 ± 51.23
Agastache	50.72 ± 0.12	37.33 ± 0.04	0.77 ± 0.02	2136.31 ± 18.94
Hawthorn	50.28 ± 0.42	32.16 ± 0.22	0.32 ± 0.03	1660.72 ± 22.31
Pomegranate	39.35 ± 0.31	48.88 ± 0.27	0.66 ± 0.02	1271.72 ± 3.84
Pepper	37.18 ± 0.23	28.22 ± 0.04	0.22 ± 0.02	1140.29 ± 13.61
Sunflower	19.72 ± 0.53	26.44 ± 0.07	0.21 ± 0.01	1018.65 ± 6.47
Rape	20.17 ± 0.23	21.22 ± 0.04	0.20 ± 0.01	985.20 ± 14.31
Vitex	21.46 ± 0.31	22.30 ± 0.07	0.12 ± 0.02	935.05 ± 6.60
Loquat	12.91 ± 0.12	21.40 ± 0.31	0.17 ± 0.03	687.61 ± 4.99

Data represent the mean of triplicate readings ± standard deviations (SD). GAE: gallic acid. QE: quercetin.

**Table 6 molecules-26-02778-t006:** Correlation analysis between indicators of Chinese honeys.

	Total Phenols	Total Flavonoids	Radical Scavenging Activity	Total Antioxidant Capacity	Inhibition Diameters	H_2_O_2_	Chroma Value
Total phenols	1.000						
Total flavonoids	0.509	1.000					
Radical scavenging activity	0.342	0.846 **	1.000				
Total antioxidant capacity	0.958 ***	0.462	0.428	1.000			
Inhibition diameters	0.795 **	0.666	0.653	0.853 **	1.000		
H_2_O_2_	0.525	0.263	0.065	0.466	0.535	1.000	
Chroma value	0.948 ***	0.577	0.507	0.940 ***	0.746 *	0.337	1.000

* means *p* < 0.05. ** means *p* < 0.01. *** means *p* < 0.001.

**Table 7 molecules-26-02778-t007:** HS-GC-IMS integration parameters of volatile compounds in tested honey samples.

Count	Compound	CAS #	Formula	MW	RI	Rt (sec)	Dt (a.u.)	Comment
1	Citral	C5392405	C_10_H_16_O	152.2	1482.3	1033.197	1.19082	
2	2,3-diethyl-5-methylpyrazine	C18138040	C_9_H_14_N_2_	150.2	1164.3	580.939	1.27668	
3	*Cis*-3,7-dimethyl-octa-2,6-dien-1-ol	C624157	C_10_H_18_O	154.3	1237.4	684.886	1.21783	
4	Citronellal-M	C106230	C_10_H_18_O	154.3	1159.6	574.247	1.34073	Monomer
5	2-Ethyl-3,5-dimethylpyrazine	C13925070	C_8_H_12_N_2_	136.2	1111.3	505.545	1.21609	
6	Methyl octanoate-D	C111115	C_9_H_18_O_2_	158.2	1110.1	503.76	1.94657	Dimer
7	Citronellal-D	C106230	C_10_H_18_O	154.3	1157.2	570.715	1.84501	Dimer
8	Linalool oxide-M	C60047178	C_10_H_18_O_2_	170.3	1069.3	445.773	1.25933	Monomer
9	Linalool oxide-D	C60047178	C_10_H_18_O_2_	170.3	1068.3	444.363	1.81632	Dimer
10	*Trans*-linalool oxide-M	C34995772	C_10_H_18_O_2_	170.3	1090.1	475.387	1.26101	Monomer
11	*Trans*-linalool oxide-D	C34995772	C_10_H_18_O_2_	170.3	1085.8	469.182	1.81801	Dimer
12	1-phenylethanol-M	C98862	C_8_H_8_O	120.2	1060.4	433.081	1.18844	Monomer
13	1-phenylethanol-D	C98862	C_8_H_8_O	120.2	1060.6	433.363	1.56989	Dimer
14	(*Z*)-2-octenal-M	C20664464	C_8_H_14_O	126.2	1050	418.281	1.32378	Monomer
15	(*Z*)-2-octenal-D	C20664464	C_8_H_14_O	126.2	1046.1	412.758	1.73993	Dimer
16	(*E*)-3-Octen-2-one-M	C18402829	C_8_H_14_O	126.2	1027.8	386.771	1.32197	Monomer
17	(*E*)-3-Octen-2-one-D	C18402829	C_8_H_14_O	126.2	1029	388.395	1.73811	Dimer
18	Benzene acetaldehyde-M	C122781	C_8_H_8_O	120.2	1040.6	404.962	1.25837	Monomer
19	Octanal-M	C124130	C_8_H_16_O	128.2	1005.5	354.937	1.41465	Monomer
20	Benzaldehyde-M	C100527	C_7_H_6_O	106.1	958.7	310.758	1.14933	Monomer
21	Benzaldehyde-D	C100527	C_7_H_6_O	106.1	959.5	311.412	1.47185	Dimer
22	(*E*,*E*)-2,4-Octadienal-M	C30361285	C_8_H_12_O	124.2	1124.2	523.896	1.26822	Monomer
23	(*E*,*E*)-2,4-Octadienal-D	C30361285	C_8_H_12_O	124.2	1123	522.193	1.77926	Dimer
24	Benzene acetaldehyde-D	C122781	C_8_H_8_O	120.2	1040	404.118	1.54104	Dimer
25	Heptanol	C53535334	C_7_H_16_O	116.2	976.8	325.815	1.39978	
26	Methyl octanoate-M	C111115	C_9_H_18_O_2_	158.2	1109.8	503.387	1.48176	Monomer
27	3-Methylpentanoic acid	C105431	C_6_H_12_O_2_	116.2	958.6	310.64	1.59685	
28	Oct-1-en-3-ol	C3391864	C_8_H_16_O	128.2	983.6	331.468	1.15686	
29	Octanal-D	C124130	C_8_H_16_O	128.2	1005.8	355.465	1.82448	Dimer
30	2-Acetylfuran-M	C1192627	C_6_H_6_O_2_	110.1	911.8	271.782	1.11613	Monomer
31	Heptanal-M	C111717	C_7_H_14_O	114.2	900.1	262.034	1.33289	Monomer
32	Heptanal-D	C111717	C_7_H_14_O	114.2	900.9	262.644	1.69957	Dimer
33	Cyclohexanone-M	C108941	C_6_H_10_O	98.1	894.8	257.566	1.15362	Monomer
34	Cyclohexanone-D	C108941	C_6_H_10_O	98.1	894.8	257.566	1.45511	Dimer
35	Furfural-M	C98011	C_5_H_4_O_2_	96.1	829.2	221.417	1.08191	Monomer
36	Furfural-D	C98011	C_5_H_4_O_2_	96.1	828	220.808	1.33289	Dimer
37	Hexanal-M	C66251	C_6_H_12_O	100.2	793	201.921	1.26118	Monomer
38	Hexanal-D	C66251	C_6_H_12_O	100.2	793	201.921	1.56105	Dimer
39	2,3-Butanediol	C513859	C_4_H_10_O_2_	90.1	781.5	196.082	1.35718	
40	(*Z*)-2-Heptenal-M	C57266861	C_7_H_12_O	112.2	945.6	299.871	1.21477	Monomer
41	(*Z*)-2-Heptenal-D	C57266861	C_7_H_12_O	112.2	946.8	300.885	1.67467	Dimer
42	3-methylbutan-1-ol	C123513	C_5_H_12_O	88.1	733.1	176.855	1.49745	
43	3-hydroxybutan-2-one	C513860	C_4_H_8_O_2_	88.1	710.5	167.878	1.33205	
44	Methyl butyrate	C623427	C_5_H_10_O_2_	102.1	694.5	161.557	1.43472	
45	3-methylbutanal	C590863	C_5_H_10_O	86.1	647.9	147.532	1.1958	
46	2-methyl-1-propanol	C78831	C_4_H_10_O	74.1	620.7	139.725	1.17089	
47	Butanal	C123728	C_4_H_8_O	72.1	556.7	121.362	1.27974	
48	2-Butanone	C78933	C_4_H_8_O	72.1	586.5	129.893	1.24377	
49	1-propene-3-methylthio	C10152768	C_4_H_8_S	88.2	682.2	157.364	1.0399	
50	Ethyl Acetate	C141786	C_4_H_8_O_2_	88.1	604.6	135.098	1.33325	
51	2-methylbutan-1-ol	C137326	C_5_H_12_O	88.1	747.7	182.666	1.47347	
52	2-methyl-2-butenal	C1115113	C_5_H_8_O	84.1	744.4	181.365	1.42826	
53	5-methylfurfural	C620020	C_6_H_6_O_2_	110.1	967.1	317.751	1.1267	
54	2-heptanone	C110430	C_7_H_14_O	114.2	890.9	254.717	1.25824	
55	Ethyl 2-methylbutyrate	C7452791	C_7_H_14_O_2_	130.2	843.4	229.121	1.65051	
56	2-Acetylfuran-D	C1192627	C_6_H_6_O_2_	110.1	912.8	272.586	1.44357	Dimer
57	Ethyl phenylacetate	C101973	C_10_H_12_O_2_	164.2	1239.1	687.359	1.30522	
58	3-Methyl-3-buten-1-ol	C763326	C_5_H_10_O	86.1	723.7	173.145	1.24758	
59	γ-Octalactone	C104507	C_8_H_14_O_2_	142.2	1279	744.06	1.34069	
60	Citronellol	C106229	C_10_H_20_O	156.3	1238.6	686.629	1.36257	

## Data Availability

The data presented in this study are available within the article.

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
