# Peer review of "Chemical Analyses and Antimicrobial Activity of Nine Kinds of Unifloral Chinese Honeys Compared to Manuka Honey (12+ and 20+)"

_molecules, 2021, doi:10.3390/molecules26092778_

Round 1
Reviewer 1 Report
The revised manuscript entitled “Three Kinds of Chinese Honeys Have Superior Antibacterial Activity in Comparison with Manuka Honey” (ID:molecules-1140685) could be accepted after the following minor revision:
Line 124. Correct the “inhibitied” to “inhibited”
Line 180, in the footnote “W/V “of the Table 4 should be written in lower case letters, i.e. w/v
Author Response
Point 1: Line 124. Correct the “inhibitied” to “inhibited”
Response 1: The modification has been completed here.
Point 2: Line 180, in the footnote “W/V “of the Table 4 should be written in lower case letters, i.e. w/v
Response 2: The modification has been completed here.
Reviewer 2 Report
Dear Authors,
Your manuscript has appropriately been improved and now it would be suitable to be published with two necessary amendments:
1: In line 135 it is shown 11 honeys........; On the other hand in table 3 they are 10 honeys. Please, remember that 9 honeys are analyzed!!!
2: Conclusions: It must be said that the Chinese honeys were compared with Manuka honeys. Further more this fact was the purpose of the research.
Author Response
Point 1: In line 135 it is shown 11 honeys........; On the other hand in table 3 they are 10 honeys. Please, remember that 9 honeys are analyzed!!!
Response 1: Thank you for your reminder. The modification has been completed here.
Point 2: Conclusions: It must be said that the Chinese honeys were compared with Manuka honeys. Further more this fact was the purpose of the research.
Response 2: The modification has been completed here. We have added the description for comparison with Manuka honeys.
This manuscript is a resubmission of an earlier submission. The following is a list of the peer review reports and author responses from that submission.
Round 1
Reviewer 1 Report
Dear authors,
The manuscript with reference molecules-1140685 entitled “Three Kinds of Chinese Honeys Have Superior Antibacterial Activity in Comparison with Manuka Honey” is, in my opinion, an interesting work with scientific quality. As more, very recent bibliographic cites are used. However, at present the manuscript requires several changes and needs some revision. The following suggestions would improve the understanding of the manuscript:
- In my personal opinion the title shows the conclusion of the work in a drastic way. It would be better “Antibacterial activity of unifloral Chinese honeys compared to Manuka honey” or something similar.
- Authors say in the introduction that 36 kinds of monofloral Chinese honeys were analysed but just 9 (Vitex, Pepper, Hawthorn, Fennel, Sunflower, Rape, Loquat, Agastache and Pomegranate) are presented in results. Please explain it.
- Lines 21-22: A general statement like this is not a result derived from the present work. It would be said “The analysed dark honeys had better antibacterial and antioxidant activities”.
- Lines 25-26: Authors set that “Ethyl 2-methylbutyrate and 3-methylpentanoic acid were the unique ingredients of agastache honey”. The term ingredient means something added, then it is more appropriated to say compounds.
- Line 37: Please, do not use dramatic expression as “people´s panic” is. It would sound better people´s concern.
- Line 38: Not drugs but remedies.
- Line 41: Please rewrite, significant and significantly are redundant.
- Line 75: Not aroma ingredients but aroma compounds
- Lines 97-98: Table 1 does not show the 11 honey samples with bacteriostatic ring larger than 11 mm in diameter.
- Lines 110-111: Authors state that “The antibacterial activity of monofloral honey is usually better than that of multifloral honey [4].”. Be careful, the reference number 4 does not say this statement. Please check it.
- Line 132: Not inhibitedit but inhibited.
- Line 135: Not ingredients but compounds.
- Lines 136-138: “It is speculated that fennel honey may contain substances with excellent 136 bactericidal ability. Fennel honey, agastache honey and pomegranate honey were three kinds of Chinese honeys with excellent antibacterial effect.”. These sentences must be deleted because they are speculations.
- Table 3: Please, give definition of MIC50 and MIC90 under the table.
- Line 206: Please, give references for “normal range”.
- Table 6: Values of HMF for Manuka honeys are quite high in comparison with the rest of the honeys. Those values need some explanation. Also, the water content for many of the samples is over 20%. This percentage will lead to fermentation of those honeys. The European normative set the limit for this parameter in 20%. Those results need some explanation.
- Line 249: Not ingredient but compound.
- Line 253: Which variables were studied under PCA? Only aroma compounds? Please provide the complete statistical analysis as additional information. Why Manuka 12+ is showed as separated group and not Manuka 20+? This is not clear.
- Material and methods: Please, explain the real studied samples and the reason for rejecting most of them.
- Line 308: Not density but concentration
- Lines 322-323: Why these times are selected?
- Line 330: Please give reference of the method for H2O2 quantitative analysis or explain it.
- Line 371: What does mean Rutin standard?
- Line 404: In general, conclusions can be improved showing a synthesis of the quantitative results.
- Line 412: Not ingredient but compound.
Author Response
Thank you for your comments concerning our manuscript entitled “Three Kinds of Chinese Honeys Have Superior Antibacterial Activity in Comparison with Manuka Honey” (ID:molecules-1140685). Those comments are all valuable and very helpful for revising and improving our paper. We have studied comments carefully and have made correction which we hope meet with approval. Revised portion are marked in red in the paper. The responds to your comments are as flowing:
Point 1: In my personal opinion the title shows the conclusion of the work in a drastic way. It would be better “Antibacterial activity of unifloral Chinese honeys compared to Manuka honey” or something similar.
Response 1: Thank you. The title has been revised according to your suggestion.
Point 2: Authors say in the introduction that 36 kinds of monofloral Chinese honeys were analysed but just 9 (Vitex, Pepper, Hawthorn, Fennel, Sunflower, Rape, Loquat, Agastache and Pomegranate) are presented in results. Please explain it.
Response 2: We first carried out antibacterial test on 36 kinds of monofloral Chinese honeys, and selected 9 honeys with excellent antibacterial properties (bacteriostatic ring larger than 11 mm in diameter). Other unifloral honey varieties with poor antibacterial ability have been eliminated. Subsequent research was carried out on the 9 selected honeys. We added an explanation in the text.
Point 3: Lines 21-22: A general statement like this is not a result derived from the present work. It would be said “The analysed dark honeys had better antibacterial and antioxidant activities”.
Response 3: Thank you. It has been revised according to your opinion.
Point 4: Lines 25-26: Authors set that “Ethyl 2-methylbutyrate and 3-methylpentanoic acid were the unique ingredients of agastache honey”. The term ingredient means something added, then it is more appropriated to say compounds.
Response 4: Modified according to your opinion.
Point 5: Line 37: Please, do not use dramatic expression as “people´s panic” is. It would sound better people´s concern.
Response 5: It has been revised according to your opinion.
Point 6: Line 38: Not drugs but remedies.
Response 6: Thank you. It has been revised according to your opinion.
Point 7: Line 41: Please rewrite, significant and significantly are redundant.
Response 7: This sentence has been modified.
Point 8: Line 75: Not aroma ingredients but aroma compounds.
Response 8: Thank you. It has been revised according to your opinion.
Point 9: Lines 97-98: Table 1 does not show the 11 honey samples with bacteriostatic ring larger than 11 mm in diameter.
Response 9: Sorry, it’s the table 2. We have corrected the error here.
Point 10: Lines 110-111: Authors state that “The antibacterial activity of monofloral honey is usually better than that of multifloral honey [4].”. Be careful, the reference number 4 does not say this statement. Please check it.
Response 10: Thank you very much for your reminding. A new reference has replaced the old one.
Point 11: Line 132: Not inhibitedit but inhibited.
Response 11: Thank you. It has been revised.
Point 12: Line 135: Not ingredients but compounds.
Response 12: Thank you. It has been revised.
Point 13: Lines 136-138: “It is speculated that fennel honey may contain substances with excellent 136 bactericidal ability. Fennel honey, agastache honey and pomegranate honey were three kinds of Chinese honeys with excellent antibacterial effect.”. These sentences must be deleted because they are speculations.
Response 13: We deleted these sentences from the article.
Point 14: Table 3: Please, give definition of MIC50 and MIC90 under the table.
Response 14: The definitions of MIC50 and MIC90 have been added.
Point 15: Line 206: Please, give references for “normal range”.
Response 15: Thank you. This statement was somewhat inaccurate. We deleted this sentence and explained the value of each indicator in the text.
Point 16: Table 6: Values of HMF for Manuka honeys are quite high in comparison with the rest of the honeys. Those values need some explanation. Also, the water content for many of the samples is over 20%. This percentage will lead to fermentation of those honeys. The European normative set the limit for this parameter in 20%. Those results need some explanation.
Response 16: The values of HMF for Manuka honeys were quite high in comparison with the rest of the honeys. Manuka honeys were purchased from New Zealand Comvita Ltd, which had been stored at room temperature for a longer time before sampling than other samples. In addition, the possible heating treatment in the factory may also increased the content of HMF in Manuka honeys. Chinese honey samples were all raw honey, which were directly stored in the refrigerator after being taken out from the hives. The water contents for 7 Chinese honeys were over 20%. This may be related to the production mode of honey in China. In order to pursue the yield, many Chinese honeys do not have enough time to mature in beehives. We have added relevant explanations in the manuscript.
Point 17: Line 249: Not ingredient but compound.
Response 17: It has been revised.
Point 18: Line 253: Which variables were studied under PCA? Only aroma compounds? Please provide the complete statistical analysis as additional information. Why Manuka 12+ is showed as separated group and not Manuka 20+? This is not clear.
Response 18: Thank you. Only aroma compounds were studied under PCA. The pollen percentage of Manuka 12+ (50.91 ± 1.84 %) was higher than that of Manuka 20+ (30.67 ± 0.81 %) . Manuka 12+ has higher purity. Therefore, Manuka 12+ was used as the sample for volatile components detection. We have added relevant explanations in the manuscript.
Point 19: Material and methods: Please, explain the real studied samples and the reason for rejecting most of them.
Response 19: Thank you. We have added relevant information in the article. We first carried out antibacterial test on all honey samples, and selected honeys with excellent antibacterial properties. Other honey varieties with poor antibacterial ability have been eliminated. Subsequent research was carried out on the selected honeys. Relevant explanations have been added in the manuscript.
Point 20: Line 308: Not density but concentration
Response 20: Thank you. It has been revised.
Point 21: Lines 322-323: Why these times are selected?
Response 21: Under the conditions of our test, these times were the K/2 rapid growth periods of the three bacteria. We have added explanation in the manuscript.
Point 22: Line 330: Please give reference of the method for H2O2 quantitative analysis or explain it.
Response 22: In an acidic environment, H2O2 can oxidize Fe2+ to Fe3+. Fe3+ ions combine with dye molecules to form Fe3+-dye complexes. The complexes have a maximum absorption wavelength at 560 nm or 595 nm, and the absorption value is proportional to the concentration of H2O2. Relevant explanations have been added in the manuscript.
Point 23: Line 371: What does mean Rutin standard?
Response 23: This was a writing error. “and compared to Rutin standard” has been deleted.
Point 24: Line 404: In general, conclusions can be improved showing a synthesis of the quantitative results.
Response 24: Thank you. The conclusions have been improved according to your suggestion.
Point 25: Line 412: Not ingredient but compound.
Response 25: Thank you. It has been revised.

Reviewer 2 Report
Corrections
Line 42. “ … and consequently impact the antimicrobial capacity”
Correction: “… and have consequently impact on the antimicrobial capacity.”
Line 44. “in vitro and in vivo”
Correction: Do in italic-type the above words.
Line 59. “, most honey will lose its antibacterial activity.”
Correction: …, most honey lose its antibacterial activity.
Line 84. “However, relatively few studies on the antibacterial aspect of Chinese honey.”
Is something missing? Rephrase the sentence.
Line 93. “(fungi)”
The plural is fungi the sing. is fungus
Line 97.
Correction: The names of the microorganisms have already mentioned so they could be written as follows: E. coli, C. albicans, S. aureus. Do the same for the whole paragraph and the text.
Line 112.
Correct indicates to indicating
Table 1. The name of botanists do not write in italic-type.
Tilia mandshurica Rup et Maxim
Tilia amurensis Rupr
Line 132. “to be inhibitedit”
Correct it.
Line 152. “was attributed the ..”
Correction: attributed to the…
Lines 176-179.
Inside the parenthesis the correct units should be mentioned i.e. 81.27 ± 1.31 mg/100 g GAE
Lines 184-185.
Correct ml to mL. Do the same for all the tables and text. Also correct in the text μl to μL
Line 446.
Check the reference
Author Response
Thank you for your comments concerning our manuscript entitled “Three Kinds of Chinese Honeys Have Superior Antibacterial Activity in Comparison with Manuka Honey” (ID:molecules-1140685). Those comments are all valuable and very helpful for revising and improving our paper. We have studied comments carefully and have made correction which we hope meet with approval. The English language and style of the manuscript have been improved. Revised portion are marked in red in the paper. The responds to your comments are as flowing:
Point 1: Line 42. “ … and consequently impact the antimicrobial capacity”. Correction: “… and have consequently impact on the antimicrobial capacity.”
Response 1: Modified according to your opinion.
Point 2: Line 44. “in vitro and in vivo”. Correction: Do in italic-type the above words.
Response 2: It has been italicized.
Point 3: Line 59. “, most honey will lose its antibacterial activity.”Correction: …, most honey lose its antibacterial activity.
Response 3: Modified according to your opinion.
Point 4: Line 84. “However, relatively few studies on the antibacterial aspect of Chinese honey.”Is something missing? Rephrase the sentence.
Response 4: We deleted this sentence from the article.
Point 5: Line 93. “(fungi)”. The plural is fungi the sing. is fungus.
Response 5: Modified according to your opinion.
Point 6: Line 97. Correction: The names of the microorganisms have already mentioned so they could be written as follows: E. coli, C. albicans, S. aureus. Do the same for the whole paragraph and the text.
Response 6: Modified according to your opinion.
Point 7: Line 112. Correct indicates to indicating
Response 7: Modified according to your opinion.
Point 8: Table 1. The name of botanists do not write in italic-type. Tilia mandshurica Rup et Maxim. Tilia amurensis Rupr.
Response 8: Modified according to your opinion.
Point 9: Line 132. “to be inhibitedit”. Correct it.”
Response 9: The spelling error has been corrected.
Point 10: Line 152. “was attributed the ..”Correction: attributed to the…
Response 10: Modified according to your opinion.
Point 11: Lines 176-179. Inside the parenthesis the correct units should be mentioned i.e. 81.27 ± 1.31 mg/100 g GAE
Response 11: The units have been added.
Point 12: Lines 184-185. Correct ml to mL. Do the same for all the tables and text. Also correct in the text μl to μL.
Response 12: It has been revised according to your opinion.
Point 13: Line 446. Check the reference
Response 13: We checked the references and corrected some formatting errors.
